# Lung Auscultation Using the Smartphone—Feasibility Study in Real-World Clinical Practice

**DOI:** 10.3390/s21144931

**Published:** 2021-07-20

**Authors:** Henrique Ferreira-Cardoso, Cristina Jácome, Sónia Silva, Adelina Amorim, Margarida T. Redondo, José Fontoura-Matias, Margarida Vicente-Ferreira, Pedro Vieira-Marques, José Valente, Rute Almeida, João Almeida Fonseca, Inês Azevedo

**Affiliations:** 1Faculty of Medicine, University of Porto, 4200-319 Porto, Portugal; hmgfcardoso@gmail.com (H.F.-C.); adelinamorim@gmail.com (A.A.); 2MEDCIDS—Department of Community Medicine, Health Information and Decision, Faculty of Medicine, University of Porto, 4200-450 Porto, Portugal; rutealmeida@med.up.pt (R.A.); fonseca.ja@gmail.com (J.A.F.); 3CINTESIS—Center for Health Technology and Services Research, Faculty of Medicine, University of Porto, 4200-450 Porto, Portugal; pmarques@med.up.pt; 4Department of Pediatrics, Centro Hospitalar Universitário de São João, 4200-319 Porto, Portugal; sonia.luisa.silva@chsj.min-saude.pt (S.S.); zpfmatias@gmail.com (J.F.-M.); margaridasvf@hotmail.com (M.V.-F.); iazevedo@med.up.pt (I.A.); 5Department of Pulmonology, Centro Hospitalar Universitário de São João, 4200-319 Porto, Portugal; margarida.tredondo@gmail.com; 6MEDIDA—Serviços em Medicina, Educação, Investigação, Desenvolvimento e Avaliação, LDA, 4200-386 Porto, Portugal; josecfvalente@gmail.com; 7Department of Obstetrics, Gynecology and Pediatrics, Faculty of Medicine, University of Porto, 4200-319 Porto, Portugal; 8EpiUnit, Institute of Public Health, University of Porto, 4050-091 Porto, Portugal

**Keywords:** respiratory sounds, auscultation, smartphone, mobile applications, asthma, cystic fibrosis, crackles, wheezes

## Abstract

Conventional lung auscultation is essential in the management of respiratory diseases. However, detecting adventitious sounds outside medical facilities remains challenging. We assessed the feasibility of lung auscultation using the smartphone built-in microphone in real-world clinical practice. We recruited 134 patients (median[interquartile range] 16[11–22.25]y; 54% male; 31% cystic fibrosis, 29% other respiratory diseases, 28% asthma; 12% no respiratory diseases) at the Pediatrics and Pulmonology departments of a tertiary hospital. First, clinicians performed conventional auscultation with analog stethoscopes at 4 locations (trachea, right anterior chest, right and left lung bases), and documented any adventitious sounds. Then, smartphone auscultation was recorded twice in the same four locations. The recordings (n = 1060) were classified by two annotators. Seventy-three percent of recordings had quality (obtained in 92% of the participants), with the quality proportion being higher at the trachea (82%) and in the children’s group (75%). Adventitious sounds were present in only 35% of the participants and 14% of the recordings, which may have contributed to the fair agreement between conventional and smartphone auscultation (85%; k = 0.35(95% CI 0.26–0.44)). Our results show that smartphone auscultation was feasible, but further investigation is required to improve its agreement with conventional auscultation.

## 1. Introduction

Respiratory diseases are a major public health problem, with a growing burden to healthcare systems and to society [1,2,3]. Asthma affects about 339 million people and represents one of the most common chronic pediatric diseases [1,4]. The overall burden and direct and indirect cost of uncontrolled asthma are high [5,6]. Cystic fibrosis (CF) has an incidence of 70,000 to 100,000 people worldwide and affects more and more the adult population [7,8]. The main impact and cause of death in CF is respiratory failure associated with chronic and recurrent infections [9], and its overall cost tends to grow as the disease progresses [10].

The burden and progressive deterioration of these diseases depend mostly on the occurrence of exacerbations, causing frequent unscheduled medical visits, emergency department visits and hospitalizations. This demands a closer monitoring of patients’ respiratory status [4,9,11]. Lung auscultation continues to be an essential part of clinical examination, as it does not require any special resources beyond a stethoscope [12], which remains an indispensable tool for diagnosis and management of respiratory diseases [13,14]. The presence and frequency of wheezing is key for the diagnosis, classification and management of asthma and its exacerbations [15], whereas exacerbations of CF can be identifiable by new crackles, associated with increased production of sputum, cough and dyspnea [11,16]. However, conventional lung auscultation has well-documented limitations, namely its dependence on the skill and auditory capacity of the clinician [17,18]. Although telemedicine is rapidly growing [19,20], avoiding costly, risky and possibly unnecessary clinical visits [21,22,23], the absence of an alternative to conventional auscultation remains a considerable limitation to the remote monitoring of patients. The COVID-19 pandemic has brought an even greater need for innovative technologies for assessing lung auscultation, given the quarantine-related limitations of going to healthcare settings [24], the social distancing measures, and the clinicians’ vestments, often impeditive to conduc this physical exam [25,26].

Digital auscultation can be of great interest to implement telemedicine services in the management of respiratory diseases. It consists of recording patients’ lung sounds with an electronic device and classifying/analyzing them automatically based on specific signal characteristics [27,28,29]. Digital auscultation has been shown to be of interest for the diagnosis and management of respiratory diseases, such as obstructive sleep apnea [30], chronic obstructive pulmonary disease [31,32] and asthma [33,34,35]. Kevat et al. has also shown that automated computerized recording and analysis of lung sounds could outperform conventional auscultation in patients with CF or patients presenting wheezes/ crackles [36]. Yet, digital auscultation continues to be insufficiently studied outside the medical context and facilities, for example, during the patient’s everyday life.

To contribute to answering these unmet research and clinical needs, we developed the AIRDOC mobile application (app) to support the remote monitoring of lung sounds in patients with chronic respiratory diseases [37], namely by enabling lung auscultation through a smartphone’s built-in microphone [19,37]. To the best of our knowledge, solutions similar to the one henceforth described are not currently available, making it one of the first lung auscultation solutions based solely on software and on the use of smartphone embedded sensors, avoiding the use of additional devices [37]. We believe this is a promising technology for boosting the usefulness of lung auscultation and to support telemedicine, as well as for improving the self-management of patients with chronic respiratory diseases. The app takes advantage of smartphones’ ubiquity in everyday life [14,37,38], as well as its portability, recording, storing and computation features [38]. However, initial testing to assess the feasibility of lung auscultation through the app in real-world clinical practice is still required. 

We aimed to study the feasibility of smartphone lung auscultation in clinical context, by assessing the quality of the recorded lung sounds and the capability of capturing adventitious sounds. Our secondary aims were (i) to compare the findings of smartphone lung auscultation with those of conventional auscultation, (ii) to compare the quality of the recordings and the identification of adventitious sounds in different respiratory diseases, and (iii) to create a lung sounds database to support the future development and validation of sound processing methods.

## 2. Materials and Methods

### 2.1. Study Design

An observational cross-sectional study was conducted with children and adults followed up at the Pediatrics and Pulmonology departments of the Centro Hospitalar Universitário de São João (CHUSJ), a Portuguese tertiary hospital. A convenience sample was recruited between September 2020 and April 2021. Patients were included if aged 5 years or over; were under medical follow-up at the Pediatrics or Pulmonology departments of CHUSJ; and could be integrated in 4 pre-established diagnostic groups (CF, other respiratory diseases, asthma, no respiratory diseases). Exclusion criteria included refusal to participate and patients whose health status or condition prevented a harmless collection of their lung sounds.

Patients were invited to participate by the clinicians during a scheduled medical appointment. Prior to data collection, written informed consent was obtained from all the adult patients, as well as from the legal guardians of every child between 5 to 17 years old. In children aged 14 to 17 years old, written informed assent was also obtained. Approval for this study was obtained from the ethics committee of CHUSJ/Faculty of Medicine of University of Porto (FMUP) (reference 316/20, 18 September 2020). The study reporting followed the strengthening the reporting of observational studies in epidemiology (STROBE) guidelines [39]. 

### 2.2. AIRDOC Mobile Application (App)

The AIRDOC mobile app aims to monitor and coach individuals with chronic respiratory diseases, using the smartphone and its integrated sensors. The app enables the recording of sounds, from lung auscultation or forced expiratory maneuvers, as well as to complete validated questionnaires [37]. In this study, only the lung auscultation feature was used. The app runs on iOS and on Android. Each clinician has created a profile in the app, in order to access the lung auscultation feature. This feature enables the recording of lung sounds in the 7 locations recommended by CORSA guidelines [27,28,29] or other locations, with an adjustable duration. Each recording includes information of the location and duration, being also possible to write additional information in a note field. In this study, this field was used to register the patients’ ID and any relevant occurrence during the recording (e.g., interferences). The current version of the AIRDOC app does not include any signal processing of the sound files. The AIRDOC app screens are depicted in Figure 1. 

Once the profile is created, the app can be used offline (to facilitate its use in the clinical context) and whenever an internet connection is available, the app automatically transfers the sound files to a secure server at the FMUP [40], from which the recordings can be accessed by authorized researchers.

### 2.3. Data Collection

Patients’ sex, age and height, as well as diagnostic group (CF, other respiratory diseases, asthma, no respiratory diseases), were initially documented by the recruiting clinicians (4 pediatricians, S.S., J.F.M., M.V.F., I.A., and 2 pulmonologists, A.A., M.T.R.) in a case report form (CRF). Conventional lung auscultation with analog stethoscopes (Littman Classic III or Littman Cardiology IV, 3M™ Littman ^®^, Maplewood, MN, USA) was then performed by the clinicians, who immediately registered in the CRF any positive findings, namely the presence of adventitious sounds [28]. Adventitious sounds were defined as wheezes (adventitious, continuous sound having a musical character [29]) or crackles (adventitious, discontinuous, explosive sound occurring usually during inspiration [29]). The conventional auscultation was performed with the patient at a seated upright position at 4 locations: the 3 locations described as the minimal recommended by CORSA—the trachea and the right and left posterior bases of the lungs [28]—and the additional CORSA location—right anterior chest, given the known high prevalence of adventitious sounds in the right middle lobe [41]. The selected locations can be seen in Figure 2a,b.

Smartphone lung auscultation was performed immediately after conventional auscultation, by the recruiting clinicians and by a final-year medical student (H.F.C.). During smartphone auscultations, clinicians firmly pressed the smartphone’s microphone to the patient’s skin and kept it stabilized, while the patient was breathing deeply with an open mouth [28]. Each clinician used his/her own smartphone, in the same 4 locations, using the AIRDOC app. In total, 8 different smartphones were used for lung sound recordings, all covered with a sanitized plastic case: an iPhone 12 Pro (iOS 14.4), an iPhone XR (iOS 14.2), an iPhone 8 (iOS 14.0), an iPhone 7 (iOS 14.1), a Xiaomi M9 Pro (Android 10), an iPhone 6s (iOS 13.6.1), a Huawei p10 lite (EMUI 8) and a One Plus 7 Pro (Android 10). Each auscultation location was recorded twice (8 recordings per participant) during 5–10 s, allowing each recording to include between 1 to 5 respiratory cycles, as recommended [13,27]. An example of the procedure can be found in Figure 2c. Since data collection took place during the COVID-19 pandemic, the patients and the clinicians wore a mask during all the procedures. 

### 2.4. Lung Sound Recording Classification

All lung sound recordings were initially listened to independently by two annotators, none of whom were recruiting clinicians (H.F.C., a final-year medical student and C.J., a physiotherapist/lung sound expert) using Adobe Audition^®^ CC 2020 version 13.0.11.38 (Adobe Inc.^®^, San Jose, CA, USA). The final-year medical student was trained by the lung sound expert in two online sessions using lung sounds acquired with digital stethoscope and respective spectrograms. Both annotators used high-quality headsets (Sennheiser^®^ HD 380 pro, Wedemark, Germany, and Sony^®^ WH-XB900N, Tokyo, Japan), while simultaneously viewing a sound spectrogram (using default parameters of Adobe Audition^®^). During classifications, annotators had only information of the patient ID, auscultation location, and were blinded to any data collected during the visit (participants characteristics, results of the conventional auscultation). The quality of each recording was evaluated (yes/no), following the European Respiratory Society definition: a sound in which there is minimal artefact, respiratory phases are visible and a sound of interest can be demonstrated [42]. Disagreement between the two annotators was settled by consensus. Then, only the lung sound recordings with quality (and the participants to whom these belonged) were considered for further analysis. Both annotators then classified the lung sound recordings with quality regarding the presence of identifiable adventitious sounds, such as wheezes or crackles. At this stage, when there was a disagreement between the two annotators, a third annotator (I.A., a pediatric pulmonologist, one of the recruiting clinicians) was asked to independently classify the recordings in question, using a high-quality headset (Sennheiser^®^ PXC 550, Wedemark, Germany) and the same audio software. The decision on presence/absence of adventitious sounds was then taken by majority rule.

### 2.5. Data Analysis

Descriptive statistics were used to characterize the participants regarding sex, age, age group (children/adults), height, and diagnostic group. Kolmogorov-Smirnov tests were used to assess the normality of the data. To explore the existence of differences across the 4 diagnostic groups, Chi-Square tests (sex, age group) and Kruskal-Wallis tests (height, age) were used.

The percentage of agreement between the two annotators in both classifications (quality and the presence of adventitious sounds) and their inter-rater agreement were calculated, the latter by use of Cohen’s kappa (k) and its 95% confidence intervals (95% CI). This was done for the total number of recordings and then considering separately the recordings of each auscultation location, age group and diagnostic group. The k values were interpreted as follows: 0–0.20 slight, 0.21–0.40 fair, 0.41–0.60 moderate, 0.61–0.80 substantial, and 0.81–1.0 almost perfect agreement [43]. 

After settling the disagreement, two feasibility metrics were calculated, the proportion of participants with at least one recording with quality and the proportion of participants with at least one recording with adventitious sounds. These proportions were calculated for each age group and diagnostic group. The proportion of recordings with quality and the proportion of recordings with adventitious sounds were also calculated for each auscultation location, age group and diagnostic group. Chi-Square tests, with Bonferroni correction when needed, were used to explore any differences in the proportions between groups. 

Finally, the agreement (percentage of agreement, k) between the two auscultation methods (conventional vs. smartphone) was explored, i.e., the agreement regarding presence/absence of adventitious sounds between clinicians notes and annotation through lung sound recordings. Two approaches were used: (i) for each age group and diagnostic group, participants were the units of analysis; (ii) for each location, age group and diagnostic group, recordings as units of analysis.

The statistics software used was IBM^®^ SPSS^®^ Statistics (version 27.0.1.0, Chicago, IL, USA). The level of significance was set at 0.05.

## 3. Results

### 3.1. Participants’ Characteristics

A total of 142 participants were recruited, but in some participants less or more than eight recordings were made, making a total of 1140 recordings, as presented in Figure 3. However, 80 recordings (amounting to eight participants) were lost due to: (i) failure in the transfer to the online server or corrupted files, (ii) during the recording the clinician identified a background noise or misplacement of the smartphone and documented in the CRF that the recording should be rejected, having recorded another as substitute, (iii) recordings obtained in the wrong location. A total of 134 participants (1060 recordings) were thus included in the analysis. Of these, 54% were male and 69% were children. The median [interquartile-IQR] age was 16 [11–22.25] years, whereas the median height was 1.59 [1.47–1.65] meters. There were 42 participants with CF, 39 with other respiratory diseases, 37 with asthma and 16 with no respiratory diseases. The participants’ characteristics are presented in Table 1.

The patients with no respiratory diseases were either patients with suspected respiratory diseases that were not confirmed during the appointment, or patients from appointments of the Attention Deficit Hyperactivity Disorder from the Pediatrics department. The patients with other respiratory diseases had bronchiectasis, primary ciliary dyskinesia, hypersensitivity pneumonitis, obstructive sleep apnea syndrome, recurrent pneumothorax, obliterative bronchiolitis, chronic obstructive pulmonary disease, atelectasis or rhinitis. 

Apart from age, with the participants in the asthma group being significantly younger than those with CF or other respiratory diseases, there were no differences among groups. 

### 3.2. Quality Classification Agreement

As shown in Figure 3, 1060 recordings were listened to and classified by both annotators. A global inter-rater agreement of 93% and a kappa of 0.82 were obtained. Regarding the auscultation location, the percentage of agreement ranged 90–97% and the inter-rater agreement 0.77–0.88, with the left posterior base showing the lowest kappa. The percentage of agreement was 93% in both age groups, with a slightly lower inter-rater agreement in the children’s recordings (k = 0.80 vs. adults k = 0.86). The group with other respiratory diseases had the lowest percentage of agreement (81%) and the highest inter-rater agreement (k = 0.93), when compared to the other three groups, whose range of percentage of agreement was 88–94% and inter-rater agreement was 0.73–0.85. The inter-rater agreement between the two annotators can be found in Figure 4 and in Appendix A. 

### 3.3. Proportion of Quality

The final classification on quality, showed that 123 (92%) participants had at least one lung sound recording with quality. No differences were found between age or diagnostic groups. These results can be found in Appendix A.

Regarding the total number of recordings that were listened to, 769 were considered to have quality (73%). The proportion of quality sounds found in the trachea (82%) was significantly higher than in the other three locations (65–72%). The children’s group had a significantly greater proportion of recordings with quality than the adults’ group (75% vs. 68%). All other comparisons were not statistically significant. These results can be found in Table 2 and in Appendix A.

### 3.4. Adventitious Sounds Classification Agreement

The 769 files that were considered to have quality were classified regarding the presence of adventitious sounds. A 91% agreement and an inter-rater agreement of k = 0.66 was found between annotators. Concerning the auscultation location, the right anterior chest had a slightly lower agreement (87% and k = 0.54), when compared to the other locations (91–93%, k 0.67–0.72). Although the percentage of agreement for both age groups was similar (90% and 93%), the inter-rater agreement in the adults (k = 0.75) was slightly higher than in the children (k = 0.61). Regarding the diagnostic groups, the asthma and CF groups showed a considerably lower inter-rater agreement (k = 0.38 and k = 0.41, respectively) when compared to the other two groups, whose inter-rater agreement ranged 0.63–0.87. The inter-rater agreement between the two annotators can be found in Figure 5. The results can also be found in Appendix A.

### 3.5. Proportion of Adventitious Sounds

The final classification on the presence of adventitious sounds, showed that 43 (35%) participants had at least one lung sound recording with adventitious sounds (Figure 3). No differences were found between age or diagnostic groups. These results can be found in Appendix A.

Only 108 (14%) of the recordings with quality had adventitious sounds. The proportion of adventitious sounds detected across each auscultation location and each age group was not significantly different. However, the proportion of adventitious sounds in the group with other respiratory diseases was found to be significantly higher when compared to the other three groups. The results can be found in Table 3 and Appendix A.

### 3.6. Conventional and Smartphone Lung Auscultation

Conventional auscultation identified adventitious sounds in 32 (26%) participants. Adventitious sounds were significantly less frequent in the asthma group and in the group with no respiratory diseases (both 0%), when compared to the CF group (36%) and the group with other respiratory diseases (49%). The adults’ group (41%) also showed a greater proportion of stethoscope identified adventitious sounds than the children’s group (14%). These results can be found in Appendix A. Regarding the smartphone auscultation, as mentioned above, 43 (35%) participants had at least one recording with adventitious sounds. With participants as unit of analysis, a 65% agreement and a kappa of 0.18 [0–0.36] were found between the two methods. These results can be found in Appendix A.

When considering recordings as unit of analysis, the comparison between conventional and smartphone auscultation evidenced a low kappa (0.35), despite having a high percentage of agreement (85%). Regarding the auscultation location, the percentage of agreement was similar across all four groups (82–88%), while the inter-rater agreement ranged 0.052–0.46, with the trachea having the lowest value of kappa (k = 0.052). Concerning the age group, both groups had a similar percentage of agreement (87% and 81%, respectively) and inter-rater agreement (k = 0.35 and k = 0.34). Finally, in terms of diagnostic groups, the group with other respiratory diseases had a percentage of agreement of 81% and the greatest inter-rater agreement (k = 0.53), while the CF group, with a percentage of agreement of 83%, had a close-to-zero value of k = 0.094. In the group with no respiratory diseases and in the asthma group (90% of agreement in both), the Cohen’s Kappa could not be calculated because the stethoscope’s classification had no positive findings. The results can be found in Figure 6. The results can also be found in Appendix A. 

## 4. Discussion

To the best of our knowledge, this is the first study assessing the feasibility of lung auscultation using only the smartphone’s built-in microphone in real-world clinical practice. Our results, namely the high proportion of participants with at least one recording with quality (92%) and the 35% of participants with identified adventitious sounds, demonstrate that smartphone auscultation is feasible using a simple set-up applied during medical appointments. Both the conventional and smartphone auscultations were performed without the need for any person or instrument besides the clinician, a conventional stethoscope, and the clinicians’ own smartphone, thus reinforcing the feasibility and practicality of using the AIRDOC app in the everyday clinical practice. These results can, in the future, help to achieve a further clinical acceptance and utility of digital auscultation, as the sole use of a smartphone does not require any additional device [44] or extra cost, and can be used outside the medical context and facilities.

Regarding the quality of the sounds, the inter-rater agreement was found to be similar to previous studies using conventional or modified stethoscopes and objective acoustical analysis by experts [17,45]. This, adding to the high proportion of recordings with quality, which amounted to more than two-thirds, shows that the smartphone built-in microphone can be successful in recording lung sounds, and therefore that auscultation using a smartphone is feasible. Additionally, we believe that the lack of quality in the excluded recordings was mostly attributable to outside noise or interference, misplacement of the smartphone or to poor respiratory pattern (e.g., low volume). As for the higher proportion of sounds with quality that was found in the trachea, it has previously been reported that respiratory sounds acquired at the trachea are higher in intensity and easier to capture than sounds from the chest wall [46].

The inter-rater agreement concerning the presence of adventitious sounds was also similar to what has been reported in the literature, including studies performed in more controlled settings [17,45]. Although only 14% of the recordings had an identifiable adventitious sound, the percentage of participants with at least one recording with adventitious sounds (35%) was similar to the prevalence found in the Tromsø population (28%). In this study, adventitious sounds were also manually identified, although lung sounds have been recorded with a different technology (using a microphone-coupled stethoscope) [47]. In the study by Murphy [48], in patients with no disease, asthma, COPD and interstitial pulmonary fibrosis, wheezes (4–59%) and crackles (21–100%) were found in a higher frequency, but direct comparisons are even more difficult, as lung sounds were acquired with microphones embedded in a soft foam mat and were automatically classified. For safety concerns related to the pandemic, patients were not observed in the outpatient department if they had acute exacerbations, but instead in the emergency room. This precluded inclusion of patients with the worst respiratory health status in the present study, thus decreasing the probability of finding abnormal lung sounds. Moreover, the significantly lower proportion of adventitious sounds in patients with asthma and CF, when compared to the patients with other respiratory diseases, might be explained by the fact that the COVID-19 pandemics has been linked to a decrease in the incidence of exacerbations in both asthma and CF [49,50], and therefore of identifiable adventitious sounds. In addition, individuals with controlled asthma frequently present without wheezing [15], and the occurrence of exacerbations in CF has progressively become more frequent in adults, and we included mainly adolescents and young adults (median age 20 [12.75–30.50]) [7]. Finally, although the presence of adventitious sounds in healthy individuals is well documented [48,51,52], our findings of 10% of the participants with no respiratory diseases presenting adventitious sounds might also be attributable to the fact that some of these participants were being followed in the outpatient clinic due to a possible, but still undiagnosed respiratory disease. 

The comparisons between the conventional and smartphone findings, despite having identified adventitious sounds in a similar percentage of participants (26% and 35%, respectively), have shown only some range of agreement. This may be attributed to the fact that conventional and smartphone auscultation were not classified by the same rater (recruiting clinicians vs. three annotators). Another possible factor that might have contributed to different classifications is the fact that conventional auscultation was commonly performed during one respiratory cycle per location and quick decisions were made by the clinicians, unlike the lung sound recordings, in which annotators could re-hear, play back and have more time to reach a decision. Additionally, in our study, the respiratory cycles which were heard with the analog stethoscope were never the same respiratory cycles as those recorded with the smartphone, as these two procedures were not conducted simultaneously [53,54]. To overcome this limitation in future investigations, it would be interesting to record the lung sounds and perform the conventional auscultation simultaneously, so that the same respiratory cycles were listened to. Other studies which have compared conventional auscultation with lung sound recordings (with digital stethoscopes and not with a smartphone) have documented percentages of agreement ranging from 75% to 100% [36,55], which are somewhat similar to the 65% and the 86% percentages of agreement found in our study (regarding the participants and the recordings, respectively). The inter-rater agreement found (k = 0.18 and k = 0.35) was, however, lower than in these studies (ranging from 0.44 to 0.55 [36,55]), which can be attributable to the fact that our study was conducted in a real-world clinical context and with a different technology. In addition, we also need to consider that our kappa values may be hampered by the low proportion of adventitious sounds found, as the Cohen’s kappa is prevalence-dependent [56]. Another study [38], which has focused on the detection of airflow, breath-phase onset, and respiratory rate with smartphone-acquired tracheal sounds (with a coupled microphone and not the embedded one), has shown good correlation with the findings of spirometry. These findings, although not pertaining to the identification of adventitious sounds, further highlight the feasibility, accuracy, and usefulness of smartphone lung sound acquisition. 

Our study has some limitations that need to be acknowledged. Our population is relatively small, but considering that the recruitment period overlaps with the COVID-19 pandemic scenario in Portugal, which has replaced most presential medical appointments with remote appointments, we consider it positive that so many participants and clinicians could be involved—once again reinforcing the feasibility and practicality of this technology. Due to this small-sized population, and as this is an initial feasibility study, our inferential analysis focused more on the recordings than on the participants. In future studies, with a larger population, it would be interesting to focus the scope of the inferential analysis on the participants. Another limitation is related to the different smartphones used for the recordings, whose microphone specifications were not possible to obtain and were therefore not studied as variables in this investigation. Nevertheless, we found that recording quality was slightly higher in the children’s group (75% vs. adults 68%), and we infer that this may be related to the fact that the majority of the recordings from children were acquired with higher-quality microphones (from iPhones). This factor merits exploration in future studies. Additionally, the use of face masks, due to the pandemic, made it difficult for the clinicians to demonstrate and correct the respiratory pattern during recordings, which could potentially have influenced the understanding and compliance with the respiratory pattern required; and even though there is no evidence of a reduced airflow when wearing masks, it is common sense that we generally feel more uncomfortable when breathing deep through a face mask than when having a normal breath. Despite all discussed limitations, this study has collected a great number of lung sound recordings, which will allow further studies and the development of methods of automatic lung sound analysis, as well as the improvement of the already existent, which were developed to deal with digital stethoscope’s recordings, and not with smartphones’ recordings. 

We believe that smartphone auscultation is a promising technology, both for the clinical context and for the future implementation of telemedicine and remote appointments. AIRDOC app can enable the clinician, and even the patient him/herself, with a ubiquitous wireless and practical tool to assess and monitor the respiratory status in-between medical appointments. This can be an innovative approach to early detect worsening periods and to implement timely interventions, ultimately leading to more cost-effective care.

## 5. Conclusions

The main findings suggest that lung auscultation with the different smartphone built-in microphones is feasible in the clinical context, as it can record lung sounds with quality and can successfully capture adventitious sounds. Further investigation is required to further develop this technology.

## Figures and Tables

**Figure 1 sensors-21-04931-f001:**
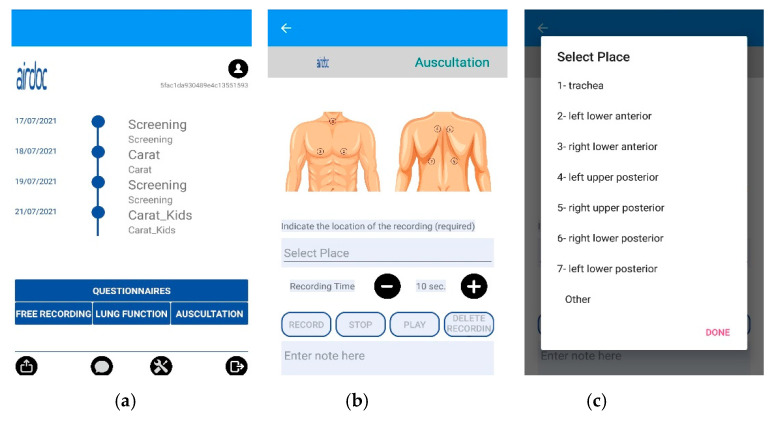
AIRDOC app screens used: (**a**) main screen; (**b**) lung auscultation feature; (**c**) location selection.

**Figure 2 sensors-21-04931-f002:**
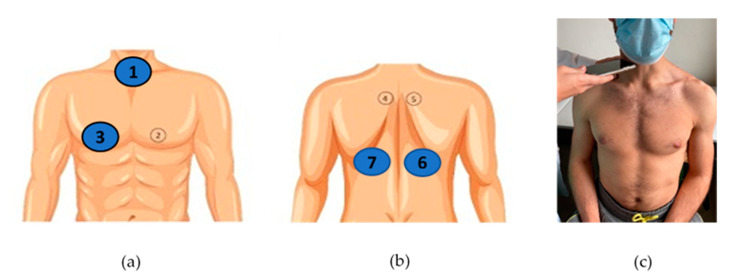
CORSA locations for lung sound recordings: (**a**) trachea (1) and right anterior chest (3); (**b**) right and left posterior bases (6 and 7); (**c**) example of a sound recording at the trachea during a medical appointment.

**Figure 3 sensors-21-04931-f003:**
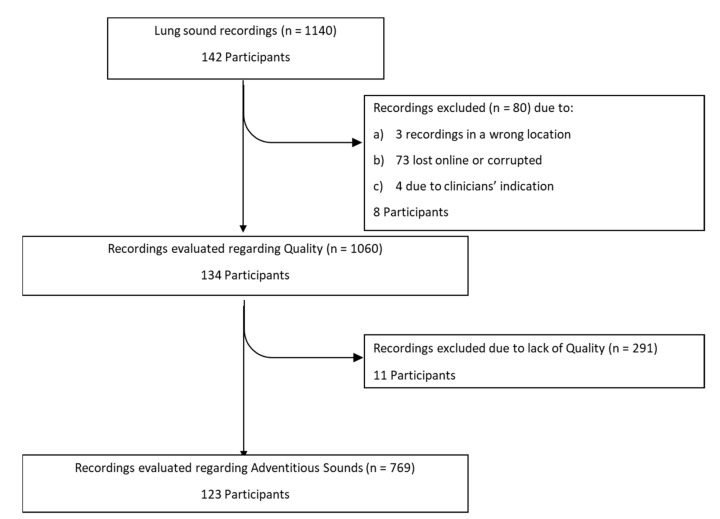
Flowchart with the number of recordings and participants throughout the study.

**Figure 4 sensors-21-04931-f004:**
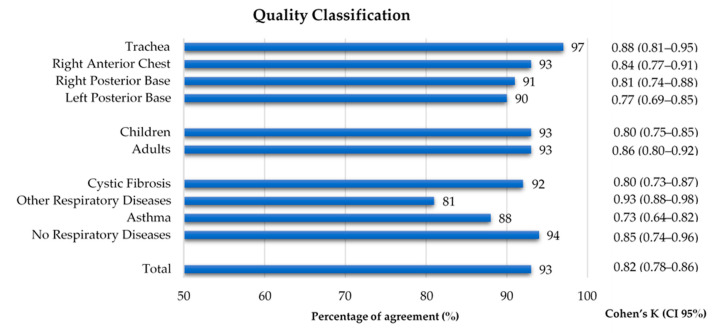
Percentage of agreement and inter-rater agreement on the quality classification of lung sound recordings by location, age group and diagnostic group.

**Figure 5 sensors-21-04931-f005:**
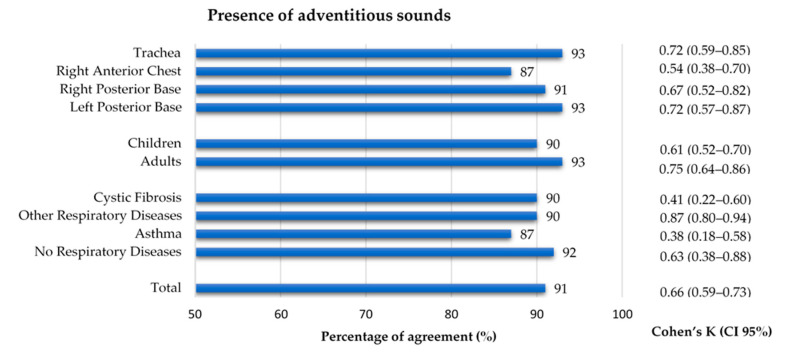
Percentage of agreement and inter-rater agreement on the presence of adventitious sounds in the lung sound recordings by location, age group and diagnostic group.

**Figure 6 sensors-21-04931-f006:**
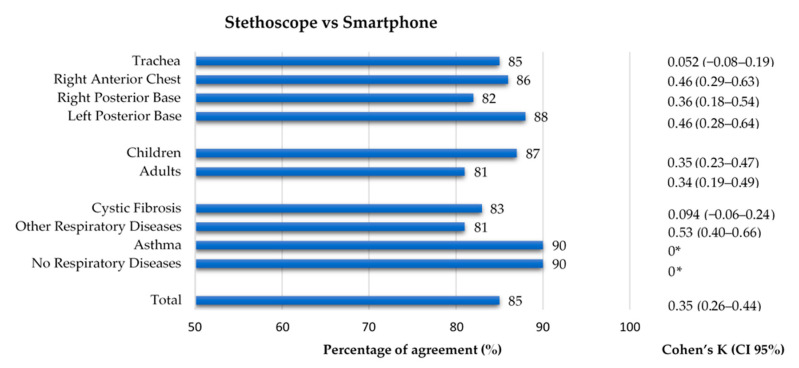
Agreement between conventional and smartphone auscultation regarding the presence of adventitious sounds by location, age group and diagnostic group. * Cohen’s Kappa was not calculated as data from conventional auscultation was constant.

**Table 1 sensors-21-04931-t001:** Participants’ characteristics (n = 134).

Characteristics	Total	Cystic Fibrosis	Other Respiratory Diseases	Asthma	No Respiratory Diseases
Subjects, n	134	42	39	37	16
Children, n (%)	92 (69)	16 (38)	26 (67)	34 (92)	16 (100)
Male, n (%)	72 (54)	19 (45)	20 (51)	22 (60)	11 (69)
Age, median [Q1–Q3] y	16 [11–22.25]	20 [12.75–30.50]	16 [12–52]	12 [9.5–16]	14 [11.25–16.75]
Height, median [Q1–Q3] m	1.59 [1.47–1.65]	1.60 [1.53–1.68]	1.57 [1.42–1.63]	1.53 [1.39–1.62]	1.63 [1.50–1.71]

Q1–Q3: interquartile range between the first and third quartiles.

**Table 2 sensors-21-04931-t002:** Proportion of sounds with quality by location, age group and diagnostic group.

		With Quality	No Quality	Proportion (%)
Location	Trachea (n = 272)	223	49	82
	Right Anterior Chest (n = 262)	189	73	72
	Right Posterior Base (n = 267)	173	94	65
	Left Posterior Base (n = 259)	184	75	71

Age group	Children (n = 710)	531	139	75
	Adults (n = 350)	238	112	68

Diagnostic group	Cystic Fibrosis (n = 354)	273	81	77
	Other Respiratory Diseases (n = 309)	221	88	72
	Asthma (n = 272)	183	89	67
	No Respiratory Diseases (n = 125)	92	33	74

	Total (n = 1060)	769	291	73

**Table 3 sensors-21-04931-t003:** Proportion of adventitious sounds by location, age group and diagnostic group.

	Adventitious Sounds	Present	Absent	Proportion (%)
Location	Trachea (n = 223)	30	193	13
	Right Anterior Chest (n = 189)	31	158	16
	Right Posterior Base (n = 173)	27	146	18
	Left Posterior Base (n = 184)	20	164	11

Age group	Children (n = 531)	69	462	13
	Adults (n = 238)	39	199	16

Diagnostic group	Cystic Fibrosis (n = 273)	18	255	7
	Other Respiratory Diseases (n = 231)	63	158	29
	Asthma (n = 183)	18	165	10
	No Respiratory Diseases (n = 92)	9	83	10

	Total (n = 769)	108	661	14

## Data Availability

The data presented in this study are available on request from the corresponding author.

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
