# Peer review of "Lung Auscultation Using the Smartphone—Feasibility Study in Real-World Clinical Practice"

_sensors, 2021, doi:10.3390/s21144931_

Round 1
Reviewer 1 Report
Dear Editor-in-chief,
The manuscript presents an interesting clinical study on the application of smartphone-based lung auscultation. The manuscript may be considered for publication if the authors consider my following comments:
Major
- Could the authors describe the inclusion/exclusion criteria of the convenience sample in Materials and Methods>Study design?
- Does the AIRDOC app implement any signal processing on the sound captured by the smartphone?
- Was smartphone auscultation done on the same day as conventional auscultation? Could the duration between the two may have played a role in inter-rater agreement between the two techniques?
- Did the same group of annotators classify the adventitious sounds for both conventional and smartphone-based auscultation? Did they classify each subject's recording from stethoscope and from smartphone simultaneously or one after another? Please clarify in the methods
- The authors should consider rephrasing the term 'quality' in sentences e.g. 'only the lung sounds with quality were considered for further analysis', 'participants had at least 280 one lung sound recording with quality'. A better term could be acceptability.
- Was quality assurance for both conventional and smartphone recordings done separately? Please clarify
- Were recordings included on analysis if both the conventional and the smartphone based auscultation was of acceptable quality? Please clarify in the methods
- The authors should include a separate paragraph in the discussion on the clinical utility of their presented application.
Author Response
Dear Sir/Madam,
We thank you for the time spent reviewing our paper and for the valuable comments. We answered each comment and made our best to address each specific recommendation. Below, we list the amendments that have been performed.
# Comment 1: Could the authors describe the inclusion/exclusion criteria of the convenience sample in Materials and Methods>Study design?
Response: Thank you for your suggestion. The eligibility criteria have now been moved to Study design subheading, as suggested (please see lines 106-111): “Patients were included if aged 5 years or over; were under medical follow-up at the Pediatrics or Pulmonology departments of CHUSJ; and could be integrated in 4 pre-established diagnostic groups (CF, other respiratory diseases, asthma, no respiratory diseases). Exclusion criteria included refusal to participate and patients whose health status or condition prevented a harmless collection of their lung sounds.”
#Comment 2: Does the AIRDOC app implement any signal processing on the sound captured by the smartphone?
Response: The current version of AIRDOC does not include any signal processing. Since it was not clear in our paper, we are thankful for your question and have now included the following sentence in the app description (please see lines 131-132): “The current version of the AIRDOC app does not include any signal processing of the sound files.”
#Comment 3: Was smartphone auscultation done on the same day as conventional auscultation? Could the duration between the two may have played a role in inter-rater agreement between the two techniques?
Response: Smartphone auscultation was done on the same day as conventional auscultation, as the participants were seen in a single medical appointment. We have now better described this both in the abstract (please see lines 29-32): “First, clinicians performed conventional auscultation with analog stethoscopes at 4 locations (trachea, right anterior chest, right and left lung bases), and documented any adventitious sounds. Then, smartphone auscultation was recorded in the same 4 locations.”; and in the Materials and Methods section (please see lines 112-113): “Patients were invited to participate by the clinicians during a scheduled medical appointment.”.
As for the duration between the two auscultation methods, first the 4 locations were heard with conventional auscultation, and then smartphone auscultation was performed in the same 4 locations. As answered above this is now clearer (please see lines 29-32): “First, clinicians performed conventional auscultation with analog stethoscopes at 4 locations (trachea, right anterior chest, right and left lung bases), and documented any adventitious sounds. Then, smartphone auscultation was recorded in the same 4 locations.”. Thus, for a specific location, the time span between conventional and smartphone auscultation was roughly 3-5 minutes (the amount of time to finish the other locations in conventional auscultation and to perform the preceding locations in smartphone auscultation). Indeed, we do believe this may have played a role in the inter-rater agreement between the two methods, as adventitious sounds often change within respiratory cycles, and therefore we have acknowledged it in the Discussion section (please see lines 457-460): “Besides, in our study, the respiratory cycles which were heard with the stethoscope were never the same respiratory cycles that were recorded with the smartphone, as these two procedures were not conducted simultaneously.”.
#Comment 4: Did the same group of annotators classify the adventitious sounds for both conventional and smartphone-based auscultation? Did they classify each subject's recording from stethoscope and from smartphone simultaneously or one after another? Please clarify in the methods
Response: Adventitious sounds heard during conventional auscultation were registered by the recruiting clinicians during the auscultation procedure itself, as they commonly do in any medical appointment. Recordings from the smartphone auscultation were classified by 2 annotators – none of whom were clinicians: a final-year medical student and a physiotherapist/ lung sound expert. A third annotator was asked to classify the smartphone recordings when there was disagreement between the 2 annotators – this third annotator is the only recruiting clinician who classified both conventional and smartphone adventitious sounds, but only in some cases (her own participants as a recruiting clinician, and some of the recordings on which there was disagreement). This has been made clearer in the Material and Methods section (please see lines: 178-179): “All lung sound recordings were initially listened to independently by two annotators, none of whom were recruiting clinicians”; and (please see lines 195-197): “At this stage, when there was a disagreement between the two annotators, a third annotator (IA, a pediatric pulmonologist, one of the recruiting clinicians) was asked to independently classify the recordings in question”.
The collection of both auscultations was not done simultaneously, but instead one after another. The classification was also done in separate moments: the conventional auscultation’s classification was done during the auscultation procedure itself, whereas the smartphone recordings’ classification was done later. This is now clearer both in abstract (please see lines 29-32): “First, clinicians performed conventional auscultation with analog stethoscopes at 4 locations (trachea, right anterior chest, right and left lung bases), documenting any adventitious sounds. Then, smartphone auscultation was recorded in the same locations.”; and in the Materials and Methods section (please see lines 159-160): “Smartphone lung auscultation was performed immediately after conventional auscultation, by the recruiting clinicians or a final-year medical student (H.F.C.).”.
#Comment 5: The authors should consider rephrasing the term 'quality' in sentences e.g. 'only the lung sounds with quality were considered for further analysis', 'participants had at least 280 one lung sound recording with quality'. A better term could be acceptability.
Response: We thank you for your suggestion. We have based our choice of the term “quality” on the lung sound nomenclature standardization advocated by the European Respiratory Society (ERS) –the most relevant scientific society in the respiratory field. Since this is an area with big nomenclature issues/dualities, we opted to follow the ERS’s definition, as not to create further nomenclature disparity.
#Comment 6: Was quality assurance for both conventional and smartphone recordings done separately? Please clarify
Response: There was no quality assessment for the conventional auscultation as it was conducted with analog stethoscopes and its findings immediately registered. This has been made clearer (please see lines 147-150): “Conventional lung auscultation with analog stethoscopes Littman Classic III or Littman Cardiology IV, 3M™ Littman ®, Maplewood, Minnesota, USA) was then performed by the clinicians, who immediately registered in the CRF any positive findings, namely the presence of adventitious sounds.”
#Comment 7: Were recordings included on analysis if both the conventional and the smartphone based auscultation was of acceptable quality? Please clarify in the methods
Response: There was no quality assessment for the conventional auscultation, as clarified in the question above. Thus, the quality analysis that determined which lung sound recordings were considered for further analysis (and therefore which participants) was done solely on the smartphone sound files.
We have altered the Materials and Methods text to make this clearer (please see lines 192-193): “Then, only the lung sound recordings with quality (and the participants to whom these belonged) were considered for further analysis.”.
#Comment 8: The authors should include a separate paragraph in the discussion on the clinical utility of their presented application.
Response: Thank you for your comment. We have now better discussed the potential of the AIRDOC app and smartphone auscultation at the end of the Discussion section. Please see lines (503-509): “We believe that smartphone auscultation is a promising technology, both for the clinical context and for the future implementation of telemedicine and remote appointments. This smartphone app can enable the clinician, and even the patient him/herself, with a ubiquitous wireless and practical tool to assess and monitor the respiratory status in-between medical appointments. This can be an innovative approach to early detect worsening periods and to implement timely interventions, ultimately leading to more cost-effective care.”.
Reviewer 2 Report
The paper is generally well written and structured. However, in my opinion,
the paper has some shortcomings in regards to presenting the results. Below I have provided numerous remarks on the text with the page number.
Comment 1: What was the difference among microphones system of different 8 smartphones which were used in this study (such as sampling rate)? (P.4. L.157-161)
Comment 2: What could be the effects of wearing a mask during the recording process.? (P.4 L.165-167)
Comment 3: How was performed the collection of data on all mentioned locations for one patient? Simultaneously or one after each other? Was the same device used In a case of simultaneously? If yes, how was the process?
Comment 4: In my opinion (Figure 6.) the chart is needed to show different percentages for stethoscope vs smartphone with two different colure. P.10 Figure 6.
comment 5: Spelling mistake in the title of Figure 6. (sthethoscope ---- > stethoscope)
Author Response
Dear Sir/Madam,
We thank you for the time spent reviewing our paper and for the valuable comments. We answered each comment and made our best to address each specific recommendation. Below, we list the amendments that have been performed.
# Comment 1: What was the difference among microphones system of different 8 smartphones which were used in this study (such as sampling rate)? (P.4. L.157-161)
Response: We thank you for your question, which is a very interesting issue and even ourselves are curious about that matter. However, we were not able to find an accurate depiction of the specifications of each of smartphone microphone, especially since even among the same smartphone brands and models these specifications often change. We have acknowledged this limitation – and its particular interest in future studies (please see lines 486-488): “Another limitation is related with the different smartphones used for the recordings, whose microphone specifications were not possible to obtain and were therefore not studied as variables in this investigation.”. We hope to continue to increase our dataset of smartphone sounds and pursue this analysis in the near future.
# Comment 2: What could be the effects of wearing a mask during the recording process.? (P.4 L.165-167)
Response: On the one hand, especially with younger participants (children), it has caused some difficulty for the clinicians to demonstrate the breathing pattern during auscultation, or even to correct the participant if he/she was not doing it properly (for example, correct the mouth aperture). Thus, we hypothesised that in some participants, lung sound recordings might have been excluded due to lack of quality because the breathing pattern was not adequate. We have now improved the discussion of this possible limitation in the Discussion section (please see lines 492-498): “Besides, the use of face masks, due to the pandemic, made it difficult for the clinicians to demonstrate and correct the respiratory pattern during recordings, which may have possibly influenced the understanding and compliance with the respiratory pattern required; and even though there is no evidence on a reduced airflow when wearing masks, it is common sense that we generally feel more uncomfortable when breathing deep though a face mask than when having a normal breath.”.
Finally, as this is an initial investigation which aims to allow for future studies (hopefully in a COVID-19 free scenario) and since one of the goals of this mobile app is to allow for remote self-auscultation (at home), the use of the facemask might be a methodological difference (and thus mention-worthy) when compared to future studies.
# Comment 3: How was performed the collection of data on all mentioned locations for one patient? Simultaneously or one after each other? Was the same device used In a case of simultaneously? If yes, how was the process?
Response: Conventional auscultation was performed on all 4 locations and only then was smartphone auscultation done on the same 4 locations, therefore sounds with the two methods were not listen/recorded simultaneously. Each clinician used his/her smartphone. We have made these two points clearer at the Material and Methods section (please see lines 159-164): “Smartphone lung auscultation was performed immediately after conventional auscultation, by the recruiting clinicians or a final-year medical student (H.F.C.). (…) Each clinician used his/her own smartphone, in the same 4 locations, using the AIRDOC app.”.
# Comment 4: In my opinion (Figure 6.) the chart is needed to show different percentages for stethoscope vs smartphone with two different colure. P.10 Figure 6.
Response: Figure 6 shows the agreement between both auscultation methods’ (conventional vs smartphone) classification of the presence of adventitious sounds, using the recordings as unit of analysis. We have now improved the figure’s caption to improve clarity (please see lines 395-396): “Agreement between conventional and smartphone auscultation regarding the presence of adventitious sounds by location, age group and diagnostic group.”. An effort to improve clarity of figures 4 and 5 and also of the analysis made was made.
please see lines 221-226): “Finally, the agreement (percentage of agreement, k) between the two auscultation methods (conventional vs smartphone) was explored, i.e., the agreement regarding presence/absence of adventitious sounds between clinicians notes and annotation through lung sound recordings. Two approaches were used: i) for each age group and diagnostic group, participants were the units of analysis; ii) for each location, age group and diagnostic group, recordings as units of analysis.”;
please see lines 378-380): “When considering recordings as unit of analysis, the comparison between conventional and smartphone auscultation evidenced a low kappa (0.35), despite having a high percentage of agreement (85%).”.
# Comment 5: Spelling mistake in the title of Figure 6. (sthethoscope ---- > stethoscope)
Response: Thank you for your comment. This has now been corrected.
Reviewer 3 Report
-The abstract section contains too much details about the study which should be presented in the Results section.
-The paper does not provide any scientific metric to measure feasibility.
-Comparison with other studies is not presented in details and based on scientific methods.
Author Response
Dear Sir/Madam,
We thank you for the time spent reviewing our paper and for the valuable comments. We answered each comment and made our best to address each specific recommendation. Below, we list the amendments that have been performed.
# Comment 1: The abstract section contains too much details about the study which should be presented in the Results section.
Response: We thank you for your comment. We have altered the abstract, namely in the part pertaining to the results as suggested (please see lines 32-37): “The recordings (n=1060) were classified by two annotators. Recordings with quality were obtained in 92% of the participants and 73% of the recordings, with the quality proportion being higher at the trachea (82%) and in the children’s group (75%). Adventitious sounds were present in only 35% of the participants and 14% of the recordings, which may have contributed to the fair agreement between conventional and smartphone auscultation (85%; k=0.35(95%CI 0.26-0.44)).”.
# Comment 2: The paper does not provide any scientific metric to measure feasibility.
Response: We thank you for your comment. We agree with you that we have failed to identify what was considered feasibility metrics by us. As we wanted to demonstrate that performing smartphone auscultation in the real-world context was feasible, we considered the proportion of participants with recordings with quality and recordings with adventitious sounds as feasibility metrics. Please find now this clarified in the Material and Methods section (please see lines 213-215): “After settling the disagreement, two feasibility metrics were calculated, the pro-portion of participants with at least one recording with quality and the proportion of participants with at least one recording with adventitious sounds.”
# Comment 3: Comparison with other studies is not presented in details and based on scientific methods.
Response: We thank you for your comment and recognize that the studies to which we compared our paper are not entirely similar. However, since this is an innovative technology, we could not find any studies comparing conventional auscultation with lung sound recordings acquired through smartphones’ embedded sensors (this is, without the use of coupled microphones, digital stethoscopes, or other instruments). Thus, we could not properly compare our results to any other studies’, forcing us to compare them with similar – but slightly different – studies.
We have, nevertheless, made an effort to highlight the most comparable aspects of these studies, namely the comparison of digital auscultation vs conventional auscultation and the accuracy and usefulness of smartphone acquired lung sounds. We provided some examples below of discussion sentences, where we compare our study with other
(please see lines 462-472): “Other studies which have compared conventional auscultation with lung sound recordings (with digital stethoscopes and not with a smartphone) have documented percentages of agreement ranging from 75% to 100%[36,55], which are somewhat similar to the 65% and the 86% percentages of agreement found in our study (regarding the participants and the recordings, respectively). The inter-rater agreement found (k=0.18 and k=0.35) was, however, lower than in these studies’ (ranging from 0.44 to 0.55[36,55]), which can be attributable to the fact that our study was conducted in a real-world clinical context and with a different technology. In addition, we also need to consider that our kappa values may be hampered by the low proportion of adventitious sounds found, as the Cohen’s kappa is prevalence-dependent[56].”.
(please see lines 472-477):” One other study[38], which has focused on the detection of airflow, breath-phase onset, and respiratory rate with smartphone-acquired tracheal sounds (with a coupled microphone and not the embedded one), has shown good correlation with the findings of spirometry. These findings, although not pertaining to the identification of adventitious sounds, further highlight the feasibility, accuracy, and usefulness of smartphone lung sound acquisition.”.
Round 2
Reviewer 1 Report
No further comments.